# A Template-Based Approach for Guiding and Refining the Development of Cinnamon-Based Phenylpropanoids as Drugs

**DOI:** 10.3390/molecules25204629

**Published:** 2020-10-11

**Authors:** Ngoc Uy Nguyen, Brendan David Stamper

**Affiliations:** Pacific University School of Pharmacy, 222 S.E. 8th Avenue #451, Hillsboro, OR 97123, USA; nguy7353@pacificu.edu

**Keywords:** phenylpropanoids, cinnamic acids, cinnamon, natural products, gene expression, drug development, DAVID

## Abstract

Background: Structure-activity relationships describe the relationship between chemical structure and biologic activity and are capable of informing deliberate structural modifications to a molecule in order enhance drug properties. Methods: Here, we present a subtle, yet unique twist on structure-activity relationships in which a collective biologic activity was measured among five cinnamon constituents with a shared phenylpropanoid template (cinnamic acid, cinnamaldehyde, chlorogenic acid, caffeic acid, and ferulic acid). This template-based approach utilized publicly available transcriptomic data through the Gene Expression Omnibus (GEO) to identify a fundamental biologic effect; in essence, a phenylpropanoid template effect. Results: The recurrent identification of cytokine-cytokine receptor interaction and neuroactive ligand receptor pathways in each individual treatment condition strongly supports the fact that changes in gene expression within these pathways is a hallmark of the phenylpropanoid template. With a template effect identified, future structural modifications can be performed in order to overcome pharmacokinetic barriers to clinical use (i.e., traditional structure-activity relationship experiments). Moreover, these modifications can be implemented with a high degree of confidence knowing that a consistent and robust template effect is likely to persist. Conclusion: We believe this template-based approach offers researchers an attractive and cost-effective means for evaluating multicomponent natural products during drug development.

## 1. Introduction

In recent years, there has been a renewed interest in the utilization and evidence-based use of natural products in conventional medicine practices [1]. Major challenges with natural product-based drug development not only include organizing the pharmacologic and toxicologic activity of individual molecules within a complex mixture, but also understanding how these multiple constituents interact with one another in potentially synergistic or antagonistic ways. Take cinnamon as an example. Resinous compounds (e.g., cinnamic acid cinnamyl acetate, and cinnamaldehyde), essential oils (e.g., eugenol, borneol, and terpinolene), as well as coumarin and procyanidins are thought to contribute to the therapeutic potential of cinnamon [2]. The structural diversity of these compounds might suggest that the potential therapeutic applications associated with cinnamon constituents varies greatly, thus making delineation of cinnamon’s pharmacologic activity quite complex.

However, despite the array of diverse molecular structures, cinnamon constituents are purported to possess remarkably similar and broad therapeutic benefits. For example, cinnamic acid derivatives have been shown to possess anti-inflammatory, antioxidant, anti-microbial, and neuroprotective activities [3,4,5]. In this study, we sought to investigate how five cinnamon constituents with a phenylpropanoid template (cinnamic acid, ferulic acid, cinnamaldehyde, caffeic acid, and chlorogenic acid) affected gene expression, and to consider whether their shared phenylpropanoid template contributed to a shared potential use in pharmacotherapy (Figure 1). It is also worth noting that the utility of natural product-based medicines is oftentimes hampered by poor pharmacokinetic properties. Even though some phenylpropanoids contain polar and ionizable groups, many are quite lipophilic and sparingly water-soluble such as certain cinnamic acid derivatives, stilbenoids, and flavonoids. In addition to potential absorption issues, complications associated with metabolism must also be considered as many natural products are susceptible to both intestinal and/or hepatic metabolism. For example, gut microbiota are capable of cleaving glycosidic bonds and transforming orally administered xenobiotics [6]. Furthermore, many natural product-based molecules also undergo metabolic modifications in the liver or kidney once absorbed [7]. The overall impact is that certain structural features and metabolic susceptibilities may result in low bioavailability and an inability to sustain suitable therapeutic serum concentrations. For this reason, many clinical trials involving a natural product-based molecule or a combination product require participants to take larger doses to achieve minimally effective concentrations. Further complicating matters is the fact that many trials end with inconclusive results [8]. Without a stable parent compound that possesses positive pharmacokinetic parameters, it can be challenging for natural product-based medicines to progress through the drug development pipeline.

Rather than evaluating individual agents for activity, this work sought to identify a robust template effect first, which could then be modified to improve pharmacokinetic parameters. Based on the fact that the phenylpropanoid template exists within all five of these cinnamic acid-like derivatives, we sought to examine to what extent the phenylpropanoid template guides pharmacologic effects. In other words, by deconstructing five molecules to their phenylpropanoid template, we might reveal important characteristics based on their foundational structural design. If common characteristics could be identified, reverse engineering could then be utilized to improve the potency and pharmacokinetic profile of the template, thus allowing researchers to simplify the complexities associated with the specific characteristics of each individual compound. Ultimately, this study hopes to present a potential approach to overcome the barriers that have prevented natural product derivatives from reaching clinical utility using cinnamon’s phenylpropanoid template as a case study.

In order to accomplish this objective, a direct comparison between cinnamon constituents was performed using a microarray dataset containing 102 traditional Chinese medicines, which was obtained from Gene Expression Omnibus (GEO) [9]. GEO is a public repository for gene expression data that contains over three million freely accessible archived samples [10]. Thus, the GEO resource allows investigators to perform both self-directed and comprehensive gene regulation analysis across a wide range of xenobiotic exposures and model systems. Once robust transcriptomic changes were determined, the Database for Annotation, Visualization and Integrated Discovery (DAVID) was utilized to coalesce these significant changes in gene expression into relevant biological pathways [11].

## 2. Results

### 2.1. Downregulated Expression Data for Cinnamon-Based Phenylpropanoids 

The 1500 most downregulated transcripts following treatment with either chlorogenic acid, cinnamaldehyde, cinnamic acid, ferulic acid or caffeic acid were significantly associated with three, four, four, four, or six KEGG pathways, respectively (Appendix A). Chlorogenic acid exposure was significantly associated with the fewest number of pathways (*n* = 3), whereas caffeic acid was significantly associated with the greatest number (*n* = 6). Interestingly, cytokine–cytokine receptor interaction and a neuroactive ligand receptor were found to be consistently downregulated across all five exposures. Within these two pathways, eight transcripts were universally identified: CCL13, CCL24, CNR1, CXCL5, GRIA1, GRM1, HGF, and THRA.

### 2.2. Upregulated Expression Data for Cinnamon-Based Phenylpropanoids

The 1500 most upregulated transcripts following treatment with either chlorogenic acid, cinnamaldehyde, cinnamic acid, ferulic acid or caffeic acid were significantly associated with 7, 12, 16, 12, 1 KEGG pathways, respectively (Appendix A). Caffeic acid exposure was significantly associated with only one pathway, whereas cinnamic acid was significantly associated with the greatest number of pathways (*n* = 16). As in the downregulated gene set, the neuroactive ligand receptor interaction pathway was consistently identified across all five treatment conditions. The fact that this pathway was identified in both the up and downregulated datasets suggests major alterations to transcript expression within this pathway are occurring in response to cinnamon-based phenylpropanoid exposure. Seven transcripts were universally identified to be upregulated in the neuroactive ligand receptor interactions pathway: CRHR2, CSH1, F2RL2, GH1, GNRHR, GRIK1, and GRIN2B. Other pathways that were common to more than one treatment condition included hematopoietic cell linage, linoleic acid metabolism, arachidonic acid metabolism, and long-term depression, which were all significantly associated with four treatment conditions (all except caffeic acid). In addition, the natural killer cell-mediated cytotoxicity pathway was significantly associated with chlorogenic acid, cinnamic acid and ferulic acid, while the vascular smooth muscle contraction pathway was significantly associated with both cinnamaldehyde and cinnamic acid (Appendix A).

## 3. Discussion

It has been proposed that cinnamon constituents have tremendous potential in drug development [12]. However, their clinical use has been limited due to a variety of factors, such as their poor pharmacokinetic parameters. A potential solution for this issue has been to combine multiple natural product-based compounds in the hopes of achieving synergistic effects (Figure 2a) [13]. Results from combined phytochemical treatments have shown promise, but many studies employing this strategy have been evaluated in simple model systems, thus making it difficult to cognize the clinical significance for human subject applications. Another approach has been to utilize drug delivery systems such as micelles to improve bioavailability (Figure 2a) [14]. However, in addition to capsule drug leakage, micelle concentration is heavily dependent on blood dilution, which can greatly influence the anticipated serum concentration of the drug [15].

Herein, we offer a unique template-based approach for developing phytochemicals as potential drugs using five phenylpropanoids found in cinnamon as an exemplar (Figure 2b). These cinnamon constituents exert remarkably similar and broad therapeutic effects, despite their varying chemical structures (Appendix A). In other words, it appears that core effects rooted in a shared structural template might exist. In this proposal, we have shown that the phenylpropanoid template related to cinnamic acid derivatives drives a consistent transcriptomic effect based on microarray data retrieved from GEO DataSeries GSE85871 [9]. The utilization of gene expression profiles in this study design offered a comprehensive view of affected downstream targets following phenylpropanoid exposure. By comparing five cinnamon constituents (Figure 1), and using gene expression profiling as the main comparative tool, the consistent identification of cytokine–cytokine receptor interaction and neuroactive ligand receptor pathways across each treatment group suggests the presence of a template effect. Now, modifications to this core template can be made in an attempt to improve the template’s pharmacokinetic characteristics.

A simple schematic was generated (Figure 3) to highlight the specific transcripts that were consistently identified in cytokine-cytokine receptor interaction and neuroactive ligand receptor pathways, the two significantly implicated KEGG pathways across all five treatment conditions (Appendix A). Cytokines, certain growth factors and other extracellular mediators, play important roles in regulating inflammatory processes via complex and sometimes contradictory interfaces [16]. The consistent identification of four downregulated cytokines (CXCL5, CCL24, CCL13, and HGF) and two upregulated hormones (CSH1 and GH1) highlight the significant and specific effects on growth and differentiation that the phenylpropanoid template is capable of eliciting. These factors within the cytokine-cytokine receptor interaction and neuroactive ligand receptor pathways that were identified are major contributors to the inflammatory response, which in turn is a critical component in the neoplastic process [17]. This provides a possible rationale for why cinnamon has been investigated for its intriguing potential as an anti-inflammatory and anticancer agent [18,19].

Not only were cytokines and growth factor transcripts identified, but the expression of receptors for these factors was also consistently altered following exposure to cinnamic acid derivatives warranting consideration as possible targets for phenylpropanoid-mediated action (Figure 3). For example, up- and down-regulation of glutamate receptor subunits (GRIA1, GRIK1, GRIN2B, GRM1) present an interesting target for phenylpropanoids. A recent in vitro study related the neuroprotective effects associated with cinnamaldehyde with the inhibition of NMDA receptors [20]. Our study also found downregulation of thyroid hormone receptor alpha (THRA) to be consistent across all five phenylpropanoid treatments, an effect also seen in rats treated with a cinnamon extract at both the mRNA and protein levels [21]. These examples further validate the template-based approach employed by this work. In other words, the phenylpropanoid template is capable of driving the observed transcriptomic changes rather than one specific molecule. This approach has also uncovered less studied targets that may play important roles in propagating the biologic effects of cinnamon constituents. For example, cannabinoid and protease-activated receptors were identified in all five cinnamon-based phenylpropanoids exposures (e.g., CNR1, F2RL3) and may hold investigatory promise considering that both signaling systems have been shown to be auspicious targets in the treatment of inflammatory disorders [22,23]. While the results from this preliminary study are promising, there are limitations to this approach. It is worth noting that the heavy reliance on gene expression profiling employed in this study may fail to accurately predict physicochemical properties that significantly impact drug formulation, half-life, and ADME parameters (absorption, distribution, metabolism and excretion).

In summary, this case study with five cinnamon constituents identified a robust and consistent template effect through the lens of transcriptomics. By diminishing the idiosyncratic pharmacologic effects and pharmacokinetic characteristics associated with individual molecules, this approach placed focus on the broad features associated with the phenylpropanoid template structure. Of additional importance is the fact that this study shines a light on valuable, yet underutilized online repositories and resources such as GEO and Data for Annotation Visualization and Integrated Discovery (DAVID). In GEO alone, there are over three million samples available to the public at no cost. Independent researchers have the ability to repurpose or collate existing sample datasets to ask novel questions that drive new discoveries, which in turn generate new hypotheses that can be tested in a laboratory or clinical setting [24]. Our study was able to identify the existence of a phenylpropanoid template effect associated with cytokine-cytokine receptor interaction and neuroactive ligand receptor pathways, from which future work will focus on structural modifications that optimize pharmacokinetic parameters. While preliminary in nature, this study has provided a justifiable launchpad for future investigations with exciting prospects in the research laboratory. We hope this new approach contributes to the development of cinnamon constituents as drugs that possess favorable pharmacokinetic properties. Future modifications to this template are not only likely to maintain pharmacologic activity inherent to the foundational structure, but also improve the chance at clinical utility in the future.

## 4. Materials and Methods

### 4.1. Data Mining

Gene expression profiles of Michigan Cancer Foundation-7 (MCF7) cells exposed to 102 different molecules used in traditional Chinese medicines were obtained through GEO DataSeries GSE85871 [9]. Gene expression profiles were extracted and compiled from 5 of the 102 molecules that were considered phenylpropanoid derivatives found in cinnamon: cinnamic acid, cinnamaldehyde, chlorogenic acid, caffeic acid, and ferulic acid. MCF7 cells exposed to a vehicle control (DMSO) were used as the control samples, and MCF7 cells exposed the aforementioned cinnamon-based phenylpropanoid derivatives were analyzed as the treatment conditions.

### 4.2. Pathway Analysis

Gene expression changes for each of the five cinnamon constituents were analyzed individually. The 1500 genes with the largest changes in expression magnitude (up or down) were uploaded into the Data for Annotation Visualization and Integrated Discovery (DAVID, version 6.7) [11,25]. The functional annotation tool was then utilized to identify enriched subsets of transcripts that were significantly associated with KEGG pathways.

### 4.3. Statistical Analysis

Significance thresholds for KEGG pathway analysis through DAVID were set at an adjusted *p*-value using a Benjamini correction (*p* < 0.05) to minimize the false discovery rate.

## Figures and Tables

**Figure 1 molecules-25-04629-f001:**
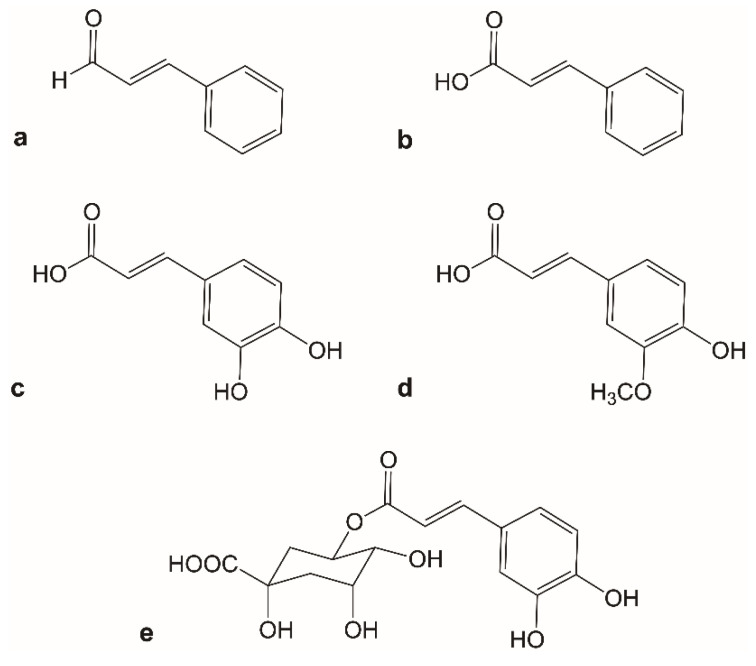
Molecular structures for (**a**) cinnamaldehyde, (**b**) cinnamic acid, (**c**) caffeic acid, (**d**) ferulic acid, and (**e**) chlorogenic acid.

**Figure 2 molecules-25-04629-f002:**
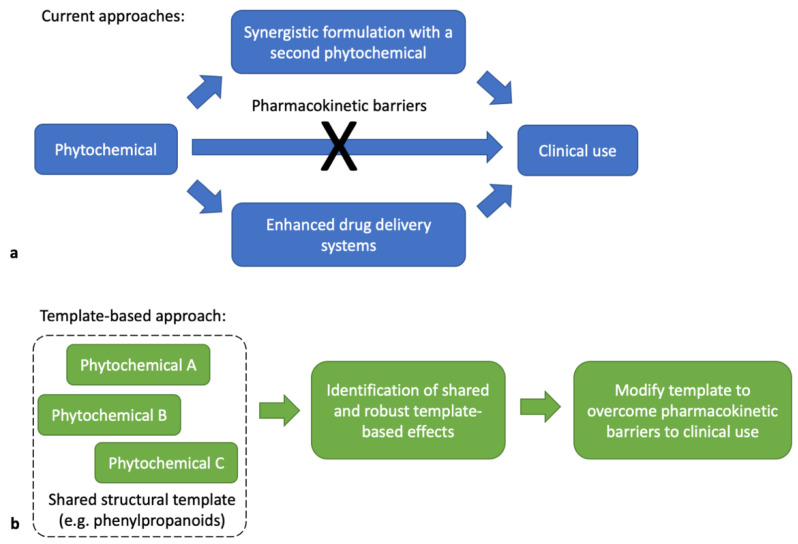
(**a**) Current and (**b**) proposed approaches to formulating phytochemicals as drugs.

**Figure 3 molecules-25-04629-f003:**
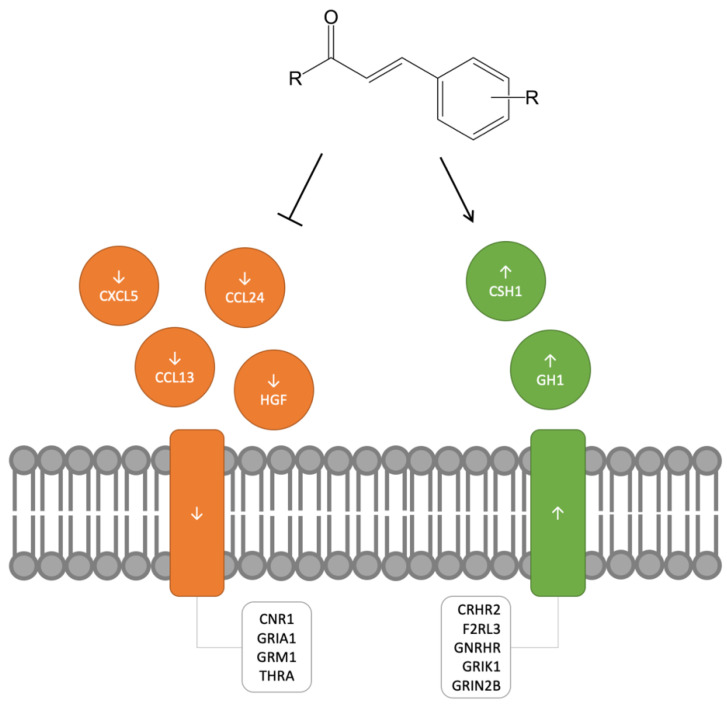
Association of fifteen transcripts identified in significantly up and downregulated KEGG pathways (cytokine-cytokine receptor interaction and neuroactive ligand-receptor interaction) and common to all five exposures to cinnamon-based phenylpropanoids.

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
