# Peer review of "A Template-Based Approach for Guiding and Refining the Development of Cinnamon-Based Phenylpropanoids as Drugs"

_molecules, 2020, doi:10.3390/molecules25204629_

Round 1

Reviewer 1 Report

The authors describe a template-based approach that makes use of transcriptomic data through the Gene Expression Omnibus (GEO) to identify key features that could facilitate development of phytochemicals into potential medicines. The article is well written and supported by reasonable transcriptomic data sets. It would however be helpful to the reader if the authors would highlight potential limitations of using this approach alone for drug screening, some of these could be failure to accurately predict compound physicochemical properties that affect critical features of drug like molecules such as formulation, half-life, absorption, distribution, metabolism and excretion. The template approach could also be used in combination with existing approaches but not as an alternative/substitute as described in some sections of the manuscript

Author Response

Point 1: It would however be helpful to the reader if the authors would highlight potential limitations of using this approach alone for drug screening, some of these could be failure to accurately predict compound physicochemical properties that affect critical features of drug like molecules such as formulation, half-life, absorption, distribution, metabolism and excretion.

Response 1: We have added a statement at the end of the second to last paragraph of the discussion to address the limitations of our study (lines 175-179): “While the results from this preliminary study are promising, there are limitations to this approach.  It is worth noting that the heavy reliance on gene expression profiling employed in this study may fail to accurately predict physicochemical properties that significantly impact drug formulation, half-life, and ADME parameters (absorption, distribution, metabolism and excretion).”

Point 2: The template approach could also be used in combination with existing approaches but not as an alternative/substitute as described in some sections of the manuscript

Response 2: This is an excellent point.  Our intent was to offer researchers a new approach to formulating phytochemicals as drugs.  Our use of the term “alternative” was not meant to express an exclusion of existing approaches.  Therefore, we have replaced our use of the term “alternative” with a less exclusionary word.  In line 129, we replaced “an alternative” was replaced with “a unique”.

Reviewer 2 Report

Manuscript molecules-958380

Entitled: A Template-Based Approach for Guiding and Refining the Development of Cinnamon-based Phenylpropanoids as Drugs

This is a well conducted scientific study, done in a thorough manner and expressed concisely. However, presented research is far from my expertise.

Author Response

Point 1: This is a well conducted scientific study, done in a thorough manner and expressed concisely. However, presented research is far from my expertise.

Response 1: Thank you for the feedback.

Reviewer 3 Report

The manuscript entitled "A template-based approach for guiding and refining the development of cinnamon-based phenylpropanoids as drugs" (Authors: Nguyen and Stamper) describe a potential novel approach to analyzed the SAR for cinnamon constituents. 

Moreover, the authors had identified the template effect for further structural modifications in order to improve the analysis. 

In order to improve the manuscript, I would like to make some suggestions to clarify some points:

  • Is it possible to explain why the authors only took into account these specific 5 cinnamon-based structures and they didn't analyze a bigger group of cinnamon-derivatives with the same core (phenylpropanoid derivatives)? 
  • Which was the control sample or control structure for these analyses?

Author Response

Point 1: Is it possible to explain why the authors only considered these specific 5 cinnamon-based structures and they did not analyze a bigger group of cinnamon-derivatives with the same core (phenylpropanoid derivatives)?

Response 1: Based on our review of the literature only 5 of the 102 molecules found in Traditional Chinese Medicine that were analyzed in GSE85871 were cinnamon-based phenylpropanoid derivatives.  This was our rational for only including these in our analyses.  We state this in the Materials and Methods section (lines 200-202): “Gene expression profiles were extracted and compiled from 5 of the 102 molecules that were considered phenylpropanoid derivatives found in cinnamon: cinnamic acid, cinnamaldehyde, chlorogenic acid, caffeic acid, and ferulic acid.”

Point 2: Which was the control sample or control structure for these analyses?

Response 2: We added a clarifying statement in the Materials and Methods section (lines 202-204): “MCF7 cells exposed to a vehicle control (DMSO) were used as the control samples, and MCF7 cells exposed the aforementioned cinnamon-based phenylpropanoid derivatives were analyzed as the treatment conditions.”

Reviewer 4 Report

The contribution of Ngoc Uy Nguyen and Brendan David Stamper entitled ‘A template-based approach for guiding and refining the development of cinnamon-based phenylpropanoids as drugs' is presenting a novel approach that should simplify, or at least be an alternative, how to test biologically active compounds of natural origin as potential drugs. Overall, the approach is interesting and is proposing an alternative to commonly used approaches (also covered in the paper). At this stage of the study I have nothing to admit, analysis, data and conclusions are correctly made and coherent. If this method will be successful or not will show only the future and additional testing of such approach in case of different classes of various natural products.

Overall, I think that the paper in the present form is suitable to be published in Molecules.

Author Response

Point 1: The contribution of Ngoc Uy Nguyen and Brendan David Stamper entitled ‘A template-based approach for guiding and refining the development of cinnamon-based phenylpropanoids as drugs' is presenting a novel approach that should simplify, or at least be an alternative, how to test biologically active compounds of natural origin as potential drugs. Overall, the approach is interesting and is proposing an alternative to commonly used approaches (also covered in the paper). At this stage of the study I have nothing to admit, analysis, data and conclusions are correctly made and coherent. If this method will be successful or not will show only the future and additional testing of such approach in case of different classes of various natural products.  Overall, I think that the paper in the present form is suitable to be published in Molecules.

Response 1: Thank you for the feedback.

Reviewer 5 Report

Nguyen et al. described a comparison study of several phenylpropanoids on global gene expression pattern using database analysis. They tried to identify the common pharmacophore to be modified for improving pharmacokinetic and bioactivity. They claimed that the use of public database gives the researchers a cost-effective way for drug design. Although the basic concept sounds interesting, their study was very preliminary. The statement was not supported by any actual experimental results. Unless there are actual synthesized compounds and its bioactivity, it is quite difficult to evaluate the value of this approach. I recommend that the author should re-organize the manuscript with experimental data of synthesized drugs.

Author Response

Point 1: Although the basic concept sounds interesting, their study was very preliminary. The statement was not supported by any actual experimental results. Unless there are actual synthesized compounds and its bioactivity, it is quite difficult to evaluate the value of this approach. I recommend that the author should re-organize the manuscript with experimental data of synthesized drugs.

Response 1: We certainly agree with this reviewer that our study was preliminary.  However, as we stated in the cover letter for our original submission: “This work represents a springboard for a ‘larger project’ involving the investigation of cinnamon-based phenylpropanoids as potential drugs, and we feel this is a good match for the Short Communications format in this special issue for Molecules.”  Per the definition under Moleclues’ scope and aim, short communications present preliminary but significant results. Therefore, we believe that our study fits these criteria.  Our goal to highlight the power and utility of accessible and comprehensive bioinformatics research tools (e.g. GEO and DAVID).  The identification of two KEGG pathways across all five treatment of cinnamon-based molecules that were statistically significant (p-benjamini < 0.05) identified a roburst gene expression profile for the phenylpropanoid template.  Additionally, our deeper analysis of the genes associated within these two pathways was further supported with primary literature. For example, the downregulation of thyroid hormone receptor alpha (THRA) was not only consistent in all five treatments, but it was also an effect that has been observed in rats treated with a cinnamon extract at both the mRNA and protein levels.  Furthermore, we strongly support the open access publication format and believe the rapid dissemination of this highly transferable work can help independent researchers generate meaningful hypotheses “at home”, especially during the pandemic where researchers are struggling to get into the lab.  We certainly don’t want to overstate the preliminary nature of our study.  We believe the inclusion of the limitations of our studies (addressed in Reviewer #1 comments; lines 175-179) helps to address Reviewer #5’s concerns.  In addition, we added a sentence in the last paragraph of the discussion (lines 191-193) to emphasize the preliminary nature of our work and the need for future wet lab experiments: “While preliminary in nature, this study has provided a justifiable launchpad for future investigations with exciting prospects in the research laboratory.”

Round 2

Reviewer 5 Report

I agree with author's claim. I see the potential value of the approach. The limitation with future wet experiments are also clearly stated. I would support the publication of the manuscript.